# Antibacterial Activity of ZnO Nanoparticles in a *Staphylococcus*-*aureus*-Infected *Galleria mellonella* Model Is Tuned by Different Apple-Derived Phytocargos

**DOI:** 10.3390/jfb14090463

**Published:** 2023-09-08

**Authors:** Catarina F. Santos, Suzana M. Andrade, Dalila Mil-Homens, M. Fátima Montemor, Marta M. Alves

**Affiliations:** 1EST Setúbal, CDP2T, Instituto Politécnico de Setúbal, Campus IPS, 2910-761 Setúbal, Portugal; 2Centro de Química Estrutural (CQE), Departamento de Engenharia Química (DEQ), Institute of Molecular Sciences, Instituto Superior Técnico, Universidade de Lisboa, Av. Rovisco Pais, 1049-001 Lisboa, Portugal; suzana.andrade@tecnico.ulisboa.pt (S.M.A.); mfmontemor@tecnico.ulisboa.pt (M.F.M.); 3Departamento de Engenharia Química (DEQ), Instituto Superior Técnico, Universidade de Lisboa, Av. Rovisco Pais, 1049-001 Lisboa, Portugal; 4iBB-Institute for Bioengineering and Biosciences, i4HB-Institute for Health and Bioeconomy, Instituto Superior Técnico, Universidade de Lisboa, Av. Rovisco Pais, 1049-001 Lisboa, Portugal; dalilamil-homens@tecnico.ulisboa.pt; 5Department of Bioengineering, Instituto Superior Técnico, Universidade de Lisboa, Av. Rovisco Pais, 1049-001 Lisboa, Portugal

**Keywords:** apple, FLIM, *Galleria mellonella*, phytocargo, *Staphylococcus aureus*, ZnO

## Abstract

This research investigates pH changes during the green synthesis of ZnO nanoparticles (NPs) and emphasises its importance in their physicochemical, antibacterial, and biological properties. Varying the synthesis pH from 8 to 12 using “*Bravo de Esmolfe*” apple extracts neither affected the morphology nor crystallinity of ZnO but impacted NP phytochemical loads. This difference is because alkaline hydrolysis of phytochemicals occurred with increasing pH, resulting in BE-ZnO with distinct phytocargos. To determine the toxicity of BE-ZnO NPs, *Galleria mellonella* was used as an alternative to non-rodent models. These assays showed no adverse effects on larvae up to a concentration of 200 mg/kg and that NPs excess was relieved by faeces and silk fibres. This was evaluated by utilising fluorescence-lifetime imaging microscopy (FLIM) to track NPs’ intrinsic fluorescence. The antibacterial efficacy against *Staphylococcus aureus* was higher for BE-ZnO_12_ than for BE-ZnO_8_; however, a different trend was attained in an *in vivo* infection model. This result may be related to NPs’ residence in larvae haemocytes, modulated by their phytocargos. This research demonstrates, for the first time, the potential of green synthesis to modulate the biosafety and antibacterial activity of NPs in an advanced *G. mellonella* infection model. These findings support future strategies to overcome antimicrobial resistance by utilizing distinct phytocargos to modulate NPs’ action over time.

## 1. Introduction

Microbial resistance to antibiotics poses serious health complications and heavy economic impacts [1,2]. In particular, the infections caused by *Staphylococcus aureus* seriously threaten human health. These bacteria can cause many diseases, such as skin infections, abscesses, impetigo, necrotising pneumonia, septicaemia, catheter-induced endocarditis, atherosclerosis, and osteomyelitis [3]. In particular, opportunistic infections in hospitals are severe, where the highly virulent antibiotic-resistant *S. aureus* is threatening public health in countries worldwide [3]. This well-known problem requires new strategies to overcome the global problem of antibiotic resistance.

The antibacterial activity of inorganic nanoparticles (NPs) has been investigated as these can trigger different antimicrobial mechanisms compared to their organic counterparts; however, their toxicity is still a concern [4]. Metal oxide NPs have garnered significant attention as new antimicrobial agents, including TiO_2_, MgO, CuO, Ag_2_O, SiO_2_, ZnO, and CaO [5,6]. Researchers have been increasingly interested in exploring the antibacterial properties of ZnO NPs, which have been found to exhibit a broad spectrum of activity against pathogenic bacteria such as *S. aureus*. Additionally, ZnO NPs are considered one of the most stable and cost-effective metal oxide NPs [7]. However, the increasing use of ZnO NPs in recent years has raised concerns about their potential toxicity. Nanoscale particles are recognised to be more reactive and toxic than their larger counterparts. This inherent danger is attributed to their small size and large surface area, which increase their reactivity and capacity to infiltrate cells, leading to amplified toxicity [8].

An *in vivo* study on *Helicoverpa armigera* larvae revealed the toxicity of green-synthesised ZnO NPs when using neem leaf extracts at doses up to 200 mg/kg [9]. Multiple *in vivo* studies on *Galleria mellonella* larvae have also reported the toxicity of conventionally synthesised ZnO NPs [10,11,12,13]. Eskin et al. [11] reported pulpal toxicity at concentrations greater than 1000 mg/kg while observing extended adult longevity at concentrations as high as 5000 mg/kg. Xu et al. [12] reported toxicity above 400 mg/kg for ZnO nanorods (NRs). The uptake of polygonal-shaped ZnO NPs by haemocytes was confirmed, and toxic effects on *G. mellonella* larvae were observed at levels above 115 mg/kg [13]. It is worth noting that pre-treating larvae with low concentrations of non-toxic ZnO NRs (≤20 mg/kg) significantly prolonged the survival of *Candida*-*albicans*-infected larvae and reduced fungal dissemination. Furthermore, adult weight increased at 100 mg/kg [11,12], indicating that concentrations in these orders of magnitude offer both antifungal and nutritional benefits. Given their antibacterial properties, ZnO NPs decreased the species richness and diversity of the *Bombyx mori* gut bacterial community and shifted their configuration to an overt microbiome [14].

Reported findings show that new strategies must be pursued to cost-effectively enhance the biosafety and effectiveness of ZnO NPs. ZnO NPs synthesised using apples, which are rich in polyphenols that possess antimicrobial and cytocompatibility properties [15,16], have the potential to offer enhanced effectiveness at lower concentrations [17]. Compared to the traditional apple varieties, *Starking* and *Golden delicious* varieties, the traditional Portuguese variety “*Bravo de Esmolfe”* is expected to exhibit significantly improved biological properties due to their higher and varying polyphenol content [18,19]. Indeed, extracts prepared from “*Bravo de Esmolfe*” were reported to have antibacterial effects, namely, against methicillin-susceptible *Staphylococcus aureus* (MSSA) and methicillin-resistant *Staphylococcus aureus* (MRSA) [16]. As such, green-synthesised ZnO with “*Bravo de Esmolfe*” phytochemicals can potentially enhance the efficacy of ZnO NPs [20], as the interaction between ZnO NPs and bioactive phytochemicals provides a promising approach to increasing the antimicrobial bioefficacy of these NPs while reducing their cytotoxicity [21,22].

No works on *in vivo* models using *G. mellonella* larvae infected with bacteria and treated with green-synthesised ZnO NPs have been reported. Therefore, this work aims to assess the bioactivity of ZnO NPs loaded with “*Bravo de Esmolfe*”-derived phytocargos on *G. mellonella*. Distinct phytocargo loadings will be achieved by adjusting the pH during synthesis. The cytotoxicity of these novel green-synthesised ZnO NPs will be tested *in vivo* on *G. mellonella*, and their antibacterial activity will be evaluated *in vitro* and in an *in vivo* infection model using *S. aureus*. The trafficking of these phytonanocarriers within *G. mellonella* will also be assessed by using fluorescence-lifetime imaging microscopy (FLIM).

## 2. Materials and Methods

Synthesis of ZnO particles: The synthesis of the ZnO NPs was made as previously described for *Starking* and *Golden delicious* apple varieties [22,23], with the final pH set to 8 or 12. Briefly, for the phytochemical extract, the common commercial apple (*Malus domestica*) variety “*Bravo Esmolfe*” was used at commercial maturity (fully ripened, Portugal). After washing, peels (0.2 g/mL) were boiled in MilliQ for 20 min, and the extract was filtered with a Whatmann No. 1 filter. After complete dissolution of zinc nitrate (Zn(NO_3_)_2_ 6H_2_O, Sigma-Aldrich, Schnelldorf, Germany) 2% (*w*/*v*), potassium hydroxide (KOH, Sigma-Aldrich, Schnelldorf, Germany) 1M solution was added to a final pH of 8 (BE-ZnO_8_) or 12 (BE-ZnO_12_). The resulting particles were collected with a nylon filter of 0.45 µm (GVS Magna, Bologna, Italy) and dried overnight in an oven (Memmert, Schwabach, Germany) at 80 °C. For comparison purposes, ZnO particles synthesised without phytochemicals at pH 12 were included [22,24].

Doping of ZnO particles with a fluorescent dye: 1 mg of rhodamine 6G (Radiant Dyes Laser & Accessories GmbH, Wermelskirchen, Germany) was mixed with 10 mg of BE-ZnO in 100 µL of water and sonicated for 5 min. The solution was then incubated at room temperature overnight. To recover the R6G-BE-ZnO particles, samples were centrifuged, washed twice with MilliQ water and dried at 40 °C [25].

Physicochemical characterisation of ZnO particles: The particles were characterised using scanning electron microscopy (SEM), X-ray photoelectron spectroscopy (XPS), UV–Vis absorption, photoluminescence (PL), and fluorescence-lifetime imaging microscopy (FLIM). Further details are provided in Appendix A.

FLIM measurements of ZnO particles obtained at pH 8 and 12: Ten microliters of the synthesised colloidal ZnO suspension (doped with R6G or not) were placed on a coverslip, and random images (80 × 80 µm^2^) were collected throughout the coverslip; regions of interest were zoomed in (up to 3 × 3 µm^2^). A 405 nm diode laser was used to study non-doped ZnO particles, whereas those doped with R6G (R6G-BE-ZnO) were analysed by using a 483 nm diode laser.

Toxicity and trafficking in *Galleria mellonella* model: The whole cycle of *Galleria mellonella* was maintained in our laboratory at 25 °C, in the darkness. Larvae were reared on a natural diet (beeswax and pollen grains), and groups of ten larvae in the final larval stage, weighing 225 ± 25 mg, were selected to be used in each experiment. *G. mellonella* survival assays were performed as described before [26] with some modifications. Larvae were injected into the hemocoel in the last left proleg, previously surface sanitised with 70% (*v*/*v*) ethanol, using a micrometre adapted to control the volume of a disposable hypodermic microsyringe. Following injection, larvae were placed in Petri dishes and stored in the dark at 37 °C. Larvae survival was observed daily, and waxworms were considered dead when they displayed no movement in response to touch. Ten larvae were used as a control and injected with phosphate-buffered saline (PBS) in all experiments.

For toxicity assays, larvae were injected with 5 µL of 2.5, 5, and 10 mg/mL ZnO particles in PBS, corresponding to 50, 100, and 200 mg/kg of ZnO particles. All injected larvae were individually examined for survival, movement, cocoon formation, and melanisation to assess the larvae health status based on the *G. mellonella* Health Index Scoring System [27]. On this scoring system, a healthy, uninfected waxworm typically scores between 9 and 10, and a dead wax worm typically scores 0. For both toxicities, larvae were assessed daily for survival up to 3 days post-treatment, being considered dead when they displayed no movement in response to touch. Kaplan–Meier survival curves were plotted using results from three independent experiments (10 larvae per group). Differences in survival rates were calculated using a log-rank (Mantel–Cox) statistical test. All analyses were performed with GraphPad Prism, version 8.0.1 software.

For trafficking assays, larvae were injected with 5 µL of 10 mg/mL ZnO NPs in PBS, corresponding to 200 mg/kg of ZnO NPs, and faeces and silk fibres were collected after 48 h for SEM and FLIM analysis.

Larvae haemolymph and haemocyte collection: *G. mellonella* larvae were injected with 5 µL of 5 mg/mL ZnO NPs suspension in PBS into the penultimate right proleg, previously sanitised with 70% ethanol. Larvae were stored in Petri dishes and incubated at 37 °C in the dark for 24 h. After incubation, three larvae for each condition were sanitised with 70% ethanol and punctured at one of the central prolegs to collect the haemolymph in an anticoagulant buffer (98 mM NaOH, 145 mM NaCl, 17 mM EDTA, and 41 mM citric acid; pH 4.5) in a 1:1 proportion. The haemolymph was centrifuged at 250× *g* for 5 min at 4 °C, washed with PBS, and pelleted haemocytes were stained using wheat germ agglutinin (WGA) conjugated with Alexa 633 (1:200; Thermo Fisher Scientific, Eugene, OR, US) for 15 min at room temperature. Then, haemocytes were centrifuged at 250× *g* for 5 min at 4 °C, washed in PBS, and resuspended in PBS for FLIM analysis.

Minimum inhibitory concentration (MIC) and minimal bactericidal concentration (MBC): The MIC was determined by a standard broth microdilution procedure recommended by the Clinical and Laboratory Standards Institute (CLSI) [28], and the minimal bactericidal concentration (MBC) was determined using a colony count procedure recommended by the Clinical and Laboratory Standards Institute (CLSI) [29]. Further details are provided in SI.

Staphylococcus aureus infection assay in *Galleria mellonella* model: For infection assay, methicillin-resistant *S. aureus* (MRSA) strain JE2 [30] was grown overnight in TSB medium at 37 °C 120 rpm. The bacterial suspension OD600 nm was measured, and the appropriate volume of cells was collected, centrifuged, and washed with PBS. Tenfold dilutions were performed to obtain 10^8^ CFU/mL. Larvae were first infected with 5 µL of the *S. aureus* JE2 suspension in the hindmost left proleg, corresponding to a 5 × 10^5^ CFU/larvae dose. This concentration corresponds to a value slightly inferior to the LD_50_ for this strain [31] and was confirmed by inoculating serial dilutions on TSA plates. After 1 h of incubation at 37 °C, larvae received 5 µL of 5 mg/mL ZnO particles suspension in PBS (corresponding to 100 mg/kg of ZnO particles) into the penultimate right proleg. After this, larvae were stored in Petri dishes and incubated at 37 °C in the dark. An untreated control group was included, infected with bacteria suspension, and treated with PBS, beside the negative control treated with PBS.

For infection assays, larvae were assessed as described in the toxicity assays.

## 3. Results and Discussion

### 3.1. Physicochemical Properties of “Bravo de Esmolfe” Phytonanocarriers

The green synthesis of ZnO nanoparticles (NPs) using apple peel extracts at pH 12 has been reported for varieties *Starking* and *Golden delicious* [22,32]. The lower total content of phenolics of these two exotic varieties, compared to “*Bravo de Esmolfe*” and their proven antimicrobial activity [16,19], makes this traditional Portuguese cultivar the basis for designing new ZnO-based antimicrobials. To create “*Bravo de Esmolfe*”-derived ZnO (BE-ZnO) with unique properties, distinct phytocargos were attained by varying the pH (pH = 8 or 12) of the green synthesis procedure (Figure 1).

The green synthesis of ZnO using “*Bravo de Esmolfe*” extracts resulted in the production of BE-ZnO crystalline NPs with round laminal-like shapes and diameters ranging from 50 to 200 nm (Figure 1a,b). The presence of diffraction rings assigned to ZnO wurtzite, namely, (1 0 0), (0 0 2), (1 0 1), (1 0 2), (1 1 0), (1 0 3), and (1 1 2), was identified in the SAED patterns of the BE-ZnO NPs (insets of Figure 1a,b) and XRD (Figure 1c). For comparison purposes, ZnO particles synthesised without phytochemicals were included [22,24]. These are crystalline micrometric flowers, composed of lamina-like structures, with sizes of approximately 3 µm (Appendix A) [22,24]. These results show that in the presence of apple phytochemicals the different pH values used during the synthesis neither impacted ZnO NPs morphology nor crystallinity. However, in their absence, microcrometric particles are formed instead [22].

Chemical analysis of the BE-ZnO NPs using EDS and XPS confirmed the presence of Zn, O, and C (Appendix A and Appendix A). A detailed XPS analysis of Zn 2p (Figure 1d) shows energies of 1021.8 and 1044.9 eV for BE-ZnO_8_ and 1021.7 and 1044.7 eV for BE-ZnO_12_, confirming the exclusive presence of ZnO [33,34,35]. These energies are attributed to Zn 2p1/2 and Zn 2p3/2 [34], indicating that Zn exists in the form of Zn(II) in both NPs [34].

A lower C and an inversely higher O content in BE-ZnO_8_ than in BE-ZnO_12_ (Appendix A) reveals that the different pH values used during ZnO NPs synthesis led to distinct phytocargos. Detailed XPS analyses of O 1s and C 1s were performed to assess the organic functional groups in each particle type (Figure 1e,f). The different shape of the O 1s spectra (Figure 1e) corroborates that distinct phytocargos are present on the ZnO surface. Three energy peaks observed for the O 1s spectrum of BE-ZnO_8_ are located at 530.5, 531.7, and 532.9 eV, and those for BE-ZnO_12_ are located at 530.8, 531.9, and 533.0 eV. Peaks at the lowest energy can be attributed to oxygen bonds in the oxide (O-Zn) [36], whereas the peaks with higher energy can be assigned to the C-O and C=O bonding of the phytocargos [37]. For the latter, binding energies attributed to chemisorbed or dissociated oxygen or OH species on the surface of the ZnO can be also considered [34]. Despite the similarity between the nature of the O 1s bonds between BE-NPs, their percentage is clearly different (Figure 1e). This observation indicates that in the synthesis made at a lower pH, the C=O contribution predominates, whereas the alkaline hydrolysis of the phytochemicals resulted in an increased C-O contribution at pH 12.

To further detail the nature of the phytocargo on BE-ZnO NPs, C 1s spectra were analysed (Figure 1f). In the BE-ZnO_8_ NPs, three peaks deconvoluted C 1s at 285.0, 286.5, and 288.8 eV, while in BE-ZnO_12_, there were four centred at 285.0, 286.5, 288.4, and 289.9 eV. Peaks with energy at 285.0 eV can originate from single or double-bounded carbons and also from adventitious carbon. At 286.5 eV, it belongs to C-O bound to alcohol alkoxy and/or ether groups, and the peak around 288 eV can be assigned to C=O from aldehyde, ketone, or ester groups. The peak exclusively detected in BE-ZnO_12_ with an energy of 289.9 eV can be assigned to carboxylic groups [38], supporting that pH increase results in phytochemicals hydrolysing into carboxylic acids.

To disclose the involved phytochemicals in the loading of BE-ZnO NPs, ATR-FTIR analysis was conducted on the “*Bravo de Esmolfo”* extract, and its spectrum was compared with those of both NPs (Appendix A). Following the synthetic process, distinct differences emerged in the phytocargos. The spectral bands in the range of 3700 to 2800 cm^−1^ can be attributed to O-H stretching. This signal is prominent in the extract due to its high water content, which hinders the detection of other existing organic compounds. The small bands evident in the powdery BE-ZnO NPs confirm the presence of organics with O-H bonds in both particles. Notably, this band is slightly more pronounced for BE-ZnO_12_ NPs than for BE-ZnO_8_ NPs, indicating higher hydrolysis of phytochemicals resulting in an increased content of O-H bonding. The disparity between the compounds in the extract and the NPs is most pronounced in the region between 1800 and 1600 cm^−1^. The broad band at 1632 cm^−1^ in the extract can be assigned to C=O and C=C, while the broad bands in the lower region at 1600 cm^−1^ indicate the prevalence of C-H bonds. Again, these changes can be attributed to the hydrolysis of functional groups present in the apple extract during the synthetic procedure at both pH values. Corroborating the increasing O-H stretching, a broad band appears at 1374 cm^−1^ in both NPs’ spectra in the region of O-H bending (Appendix A). An increase in C-O content, depicted by XPS (Figure 1e), with an increasing pH during the synthesis of NPs, is also confirmed by a larger intensity for BE-ZnO_12_ compared to BE-ZnO_8_ in the ATR-FTIR band at 1040 cm^−1^ ascribed to C-O stretching (Appendix A). While the extract displays a prominent signal between 1000 and 750 cm^−1^, associated with C=C bonds, the sharp band observed at 816 cm^−1^ in both NPs can be attributed to Zn-O stretching. This observation agrees with the existing literature on ZnO NPs synthesised with other apple extracts [22].

Although the pH change during the green synthesis did not dramatically affect the size and shape of the NPs, it had a considerable impact on the loading of phytochemicals. This result proves that distinctive phytonanocargos were achieved through the pH change during the green synthesis procedure. This agrees with a report of Abdol Aziz et al. [39], where alterations in the pH of the synthesis of ZnO using banana peel extracts rendered altered FTIR bands, and with our previous work demonstrating that starting organic compounds decomposed into simpler chemical structures during the synthesis of different ZnO particles [22].

### 3.2. Toxicity of BE-ZnO Phytocarriers

To evaluate the *in vivo* toxicity of ZnO NPs synthesised via green methods, the *Galleria mellonella* larvae model was utilised as an alternative to non-rodent models [40]. A toxicity assay with increasing concentrations of the NPs (50, 100, and 200 mg/kg of larvae) was conducted over 4 days. For all particle concentrations, a survival rate of 100% was observed (Figure 2a). This observation agrees with the range of efficacy and non-toxicity reported for non-green-synthesised ZnO NPs in *Galleria mellonella* [11,12].

Along with survival, three other parameters were monitored, namely, movement, cocoon formation, and melanisation, to assess the health status of the larvae using the *G. mellonella* Health Index Scoring System [27]. Upon visual inspection, all larvae presented a healthy status, i.e., the larvae injected with ZnO particles either without or with phytocargo had a healthy score like the ones injected with PBS (control) (Figure 2b). In contrast to the findings of Eskin et al. [11] for ZnO NPs synthesised via traditional methods, we observed no improvement in the health status of larvae following the administration of ZnO NPs synthesised via green methods nor of microparticles synthesized by traditional methods.

### 3.3. Antibacterial Activity of BE-ZnO NPs

Microbiological tests were performed to evaluate the antibacterial efficacy of these NPs against an important pathogen, methicillin-resistant *Staphylococcus aureus* (MRSA). The results showed MIC and MBC values of 313 µg/mL for the BE-ZnO_12_ particles and 625 µg/mL for the microparticles and BE-ZnO_8_ particles. The values for BE-ZnO_12_ align with those reported for ZnO NPs MICs up to 500 µg/mL, whereas those for BE-ZnO_8_ are above [7]. Despite the many reports on the antibacterial activity of green-synthesised ZnO NPs [41], only one reports ZnO synthesised with a Golden apple variety using pH 12 with antibacterial activity against Gram-positive bacteria, *S. aureus*, and also Gram-negative, *Escherichia coli* [23]. Our findings stand out, as well as a higher antibacterial efficacy for the BE-ZnO_12_ particles than for BE-ZnO_8_, suggesting that the hydrolysed apple phytochemicals are somehow more effective. These results agree with reports on ZnO NPs synthesised using orange peel extract, where increased bactericidal ability against *S. aureus* was observed from particles synthesised at pH 10 than at pH 8 [42]. Contrary to our work, no comparison was made for the antibacterial activity of ZnO particles without phytocargos [42].

To further evaluate the bacterial activity of the particles, larvae infected with *S. aureus* were injected with ZnO particles with no phytocargo and with *“Bravo de Esmolfe”* phytonanocargos. Within 24 h, the lowest survival rate was observed for the control larvae (PBS), when compared with any of the larvae injected with the ZnO particles. It is noteworthy that the survival trend changed over time. Within 48 h, BE-ZnO_12_ became more effective than BE-ZnO_8_, followed by ZnO microparticles and the control, with the highest mortality rate. Within 72 h, less than 5% of the control larvae (PBS) survived the infection, while those incubated with ZnO microparticles reached no more than 15% (Figure 3). After this period, BE-ZnO NPs resulted in a 25–30% survival rate. This is the first report of the antibacterial activity of green-synthesised ZnO NPs in an infected *Galleria mellonella* model.

While the reduced antibacterial efficacy of micrometric flowers, in comparison to BE-ZnO NPs, could be attributed to the absence of phytochemicals or variations in their morphological characteristics, the distinct bioactivities of BE-ZnO NPs may be ascribed to their specific phytocargos.

### 3.4. Particle Trafficking Inside Galleria mellonella

To better understand the difference observed between the *in vivo* response of infected *G. mellonella* to the BE-ZnO NPs, the trafficking inside the larvae was studied. As haemocytes play a crucial role in the immune response of larvae by recognising and eliminating bacterial infections, the interaction of the phytonanocargos within the haemocyte environment was investigated.

To assess the integrity of the haemocytes when in contact with the BE-ZnO NPs, the cells were stained with a bright, photostable red-fluorescent dye—Alexa 633—and monitored using FLIM. This dye will preferentially be located in the cell membrane (as depicted in Figure 4a), with a lifetime distribution in the range of 2–4 ns, obtained upon 633 nm laser excitation (as expected, no signal of this dye was detected with the 485 nm laser excitation, as shown in Figure 4d). The integrity of this membrane dye in the presence of the BE-ZnO particles was proven by the identical fluorescence spectra attained of the stained haemocytes with or without BE-ZnO NPs (Appendix A). While in the presence of the BE-ZnO_8_ NPs, the haemocytes were kept intact (Figure 4e,g); with the BE-ZnO_12_ NPs, the integrity observed 1 h after injection was lost 24 h after injection (Figure 4k,m). The fluorescence lifetime distribution for the cells in the presence of BE-ZnO_8_ (Figure 4i) shows that the signal of the haemocytes is like that of the control 24 h after the injection, whereas, with the BE-ZnO_12_ NPs (Figure 4o), the signal was similar to the control and shifted 24 h after.

These changes indicate that the signal from Alexa 633 in the membranes is more perturbed by the presence of R6G-BE-ZnO_8_ particles after 1 h. In contrast, significant changes in Alexa 633 signal occur after 24 h of R6G-BE-ZnO_12_ NP injection. This observation suggests that different interactions of the BE-ZnO NPs occurred due to their distinct phytonanocargos. Since the residence of antibacterial NPs on haemocytes [43] can be behind the increased bactericidal activity of these immune cells, we have further explored the location of the BE-ZnO NPs. A dye was coupled to the particles to minimise the background “contamination” in our signal. Rhodamine 6G (R6G), a laser dye with high brightness (strong visible absorption and high fluorescence quantum yield) and photostability [44], was used to tag BE-ZnO NPs (Appendix A).

While no signal was detected for R6G on the haemocytes without particles (Figure 4b,d), the haemocytes of larvae injected with R6G-BE-ZnO_8_ (Figure 4f,h,j) showed clear signals from the coupled dye 1 h after injection, which decreased 24 h after. For the R6G-BE-ZnO_12_ (Figure 4l,n,p), the same decrease in the signal was observed. In both cases, a broad lifetime distribution centred at 3 ns was obtained with similar intensity, indicating that the coupling between the R6G and the BE-ZnO_12_ NPs was kept. Nonetheless, the more intense and shifted lifetime distribution to shorter lifetimes observed in the case of R6G-BE-ZnO_8_ NPs after 1 h indicates that a higher internalisation is achieved for these particles. In both cases, the presence of the BE-ZnO NPs inside the haemocytes 1 h after injection was proven, as no signal for the control samples was depicted. These results agree with previous reports on ZnO NPs uptake by circulating haemocytes [43].

The uptake of both BE-ZnO NPs appears to be similar, as evidenced by their detection 1 h after injection. However, their interaction with the haemocyte membrane differs significantly. While the perturbation of Alexa 633 signals by the presence of BE-ZnO_8_ NPs after 1 h suggests higher adsorption of these particles, significant changes in Alexa 633 signals following 24 h of BE-ZnO_12_ NPs injection suggest that the latter had a longer residence time on the membrane.

The interaction of NPs with the membrane of larvae haemocytes has been previously reported for conventional ZnO [43]. However, the modulation of ZnO interactions due to phytocargo loading has never been assessed. This work found a clear relationship between phytocargo loading and NP residence in haemocytes. We assume that unhydrolysed phytochemicals in BE-ZnO_8_ favoured earlier haemocyte penetration, while hydrolysed phytochemicals in BE-ZnO_12_ increased membrane residence (Figure 1 and Figure 4).

Previous studies reported that biomolecules found in the NP corona can trigger specific recognition by cell membrane receptors, significantly affecting the adhesion properties of NPs to the membranes and their consequent uptake and internalisation rate [45]. Likewise, by changing phytocargos, the recognition and uptake of the BE-ZnO NPs by the haemocytes differ, proving that green synthesis is powerful in modulating NP perception by cell membranes.

This work provides evidence that altering the pH during ZnO synthesis in the presence of *“Bravo de Esmolfe”* extract results in unique phytocargos. These distinctive phytocargos play a significant role in influencing the antibacterial activity of BE-ZnO NPs. The antibacterial effectiveness of these NPs is closely correlated to their interaction with larvae haemocytes, i.e., BE-ZnO_8_ exhibits faster internalization, leading to swift antibacterial activity, while BE-ZnO_12_ showcases delayed internalization, correlating with a postponed antibacterial response (Figure 5).

To provide biomedical safety information regarding the clearance of BE-ZnO NPs from *G. mellonella*, the intrinsic photoluminescence properties of ZnO [46] were used to track particle internalisation. Due to their shorter residence time inside the haemocytes (Figure 4), BE-ZnO_8_ particles were selected for this study. BE-ZnO_8_ NPs’ intrinsic photoluminescence was evaluated by using FLIM, with laser excitation at λ = 405 nm. FLIM images show small, isolated entities coexisting with very few larger structures–agglomerates, and the associated fluorescence lifetime distribution (Figure 6) exhibited the prevalence of a short fluorescence lifetime (peak = 1.1 ± 0.2 ns). This value agrees with values previously ascribed to apple-derived ZnO NPs [22].

Forty-eight hours upon injection of 200 mg/kg of BE-ZnO_8_ NPs, larvae faeces and silk fibres were analysed using FLIM (Figure 7).

Both FLIM (Figure 6) and SEM images (Appendix A) depict a high heterogeneity of the faecal samples in the absence (Figure 7a) and presence (Figure 7b) of BE-ZnO_8_ NPs. In larvae either injected with or without NPs, commensal bacteria and pollen grains [47] are visible, respectively (green–red and blue objects in Figure 7a,b). Pollen grains are responsible for the fluorescence lifetime distribution with a peak of around 2 ns. In contrast, commensal bacteria account for the fluorescence lifetime distribution at longer fluorescence lifetimes (peak ≈ 3 ns; Figure 7c). In the injected larvae, some agglomerates of BE-ZnO NPs (~600 nm) can be observed in distinct parts of the faecal samples (Figure 7b), especially in the pollen grain surface (Figure 7b–i). The average fluorescence lifetime distribution (Figure 7c) associated with image Figure 7b evidence two distinct regions: shorter lifetimes (peak around 1 ns), which matches that obtained from region (i), and longer lifetimes (peak nearly 2 ns), which matches that in region (ii). The first one corresponds to pollen grain containing BE-ZnO_8_ NPs (Figure 7c) and is shorter than those obtained without the particles (Figure 7c), as the particles exert a quenching effect on the intrinsic pollen fluorescence. In contrast, the second one relates to an agglomerate of other faecal entities (Figure 7b-ii). SEM image inspection of silk fibres (Appendix A) reveals no morphological differences between the control and the larvae silk fibres treated with BE-ZnO_8_ NPs. FLIM images of larvae silk fibres upon PBS (control) or BE-ZnO_8_ NPs injection (Figure 7d,e, respectively) show that the latter already displays an intense fluorescence signal with an average lifetime peak at around 3 ns (Figure 7f). The presence of BE-ZnO_8_ NPs agglomerates of ca. 600 nm can be distinguished in the silk fibres (insets in Figure 7e).

Our data suggesting that larvae can eliminate BE-ZnO_8_ NPs through excretion is supported by reports indicating that ZnO NPs can cross the insect gut barrier [43]. This observation aligns with previous studies demonstrating that larvae excrete NPs, including Ag, CuO, and ZnO NPs, primarily through faeces [14,48]. The presence of BE-ZnO_8_ NPs in silk fibres suggests another potential pathway of excretion of the NPs, an observation supported by reported research on *Bombyx mori* that revealed higher concentrations of Zn in the silk fibre glands of individuals treated with ZnO NPs [49].

## 4. Conclusions

This research concludes that the different pH values used during the green synthesis procedure neither affected the morphology nor crystallinity of BE-ZnO NPs. This was proven by means of TEM, SAED, and XRD techniques. However, it significantly impacted the loading of phytochemicals, which resulted in BE-ZnO NPs with distinct apple phytonanocargos, as depicted by EDS, XPS and ATR-FTIR data. These latter two techniques pointed to the predominance of C=O groups over that of C-O at pH 8, whereas at pH 12, equivalent contributions of these two functional groups were attained upon the alkaline hydrolysis of the apple-derived phytochemicals.

Cytotoxicity assays with an advanced *G. mellonella* model supported the biosafety of the BE-NPs, as the larvae injected with BE-ZnO NPs at a concentration of up to 200 mg/kg had a 100% survival rate and maintained their health status. This biosafety may be due to the ability of larvae to excrete the NPs through faeces and silk fibres.

The antibacterial activity of BE-ZnO_12_ NPs was superior to that of BE-ZnO_8_ against *S. aureus*, which can be attributed to different apple-derived phytocargos. The lack of toxicity and antibacterial activity make BE-ZnO NPs a promising alternative for antibacterial applications. This conclusion was further supported by an infection larvae model, where BE-ZnO_12_ NPs were less effective than BE-ZnO_8_ 24 h after injection but became the most effective 48 h post-infection. This behaviour seems related to BE-ZnO_8_ and BE-ZnO_12_ NPs’ earlier haemocyte penetration and increased membrane residence, respectively. These results demonstrate that green synthesis is a powerful tool to modulate the perception of NPs by both bacterium and haemocyte cell membranes. Future strategies to overcome antimicrobial resistance can utilize distinct phytocargo natures to modulate their action over time and increase their efficiency.

Overall, the findings highlight the potential of the green synthesis of ZnO NPs to modulate biological responses and provide biosafe and effective antibacterials and help overcome the increasing resistance of their organic counterparts.

## Figures and Tables

**Figure 1 jfb-14-00463-f001:**
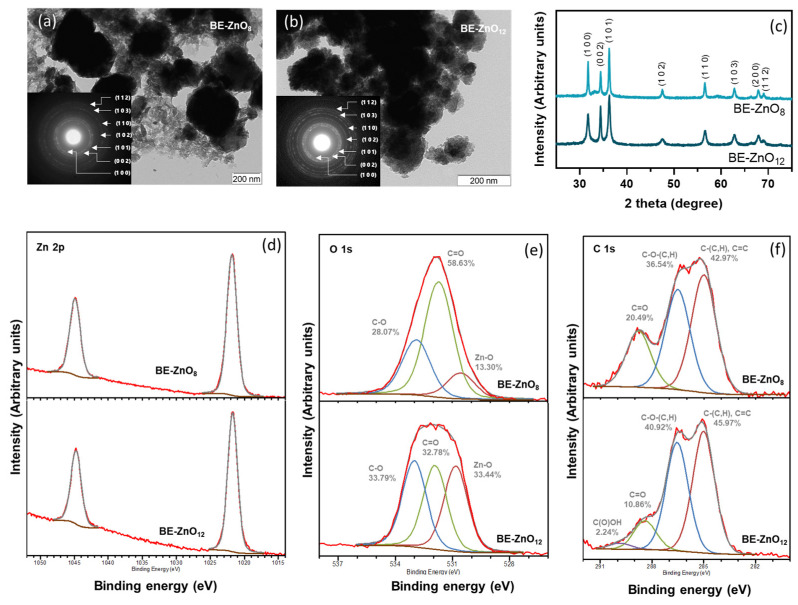
Physicochemical characterisation of BE-ZnO NPs with different phytocargo loads from “*Bravo de Esmolfe*” attained at pH 8 (BE-ZnO_8_) or pH 12 (BE-ZnO_12_): (**a**,**b**) transmission electron microscopy (TEM) images with the corresponding electron diffraction (ED) patterns as insets and (**c**) X-ray diffractograms; (**d**–**f**) X-ray photoelectron spectroscopy (XPS) of Zn 2p, O 1s, and C 1s, respectively, where black line corresponds to the experimental data and coloured lines to fitting.

**Figure 2 jfb-14-00463-f002:**
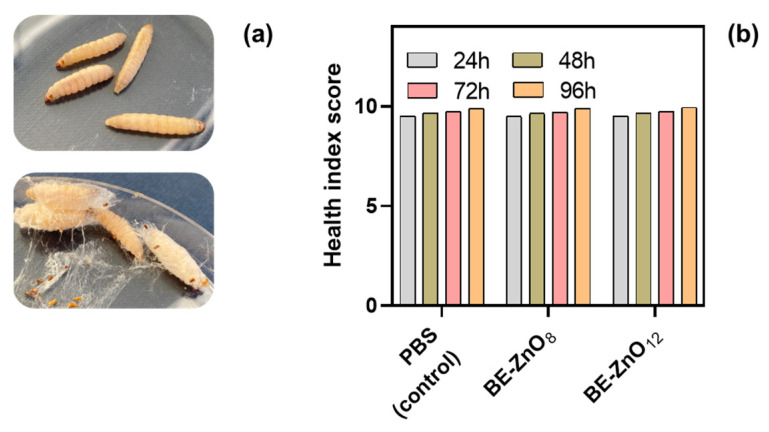
Bioactivity of ZnO particles in *Galleria mellonella*. Cytotoxicity of the particles after injection in the larvae up to 96 h: (**a**) representative optical images and (**b**) representative health index score (≥9 represents a healthy larva and a score <9 represents an affected larva) of larvae treated with 200 mg/kg of ZnO particles (no phytocargo) or with BE-ZnO NPs.

**Figure 3 jfb-14-00463-f003:**
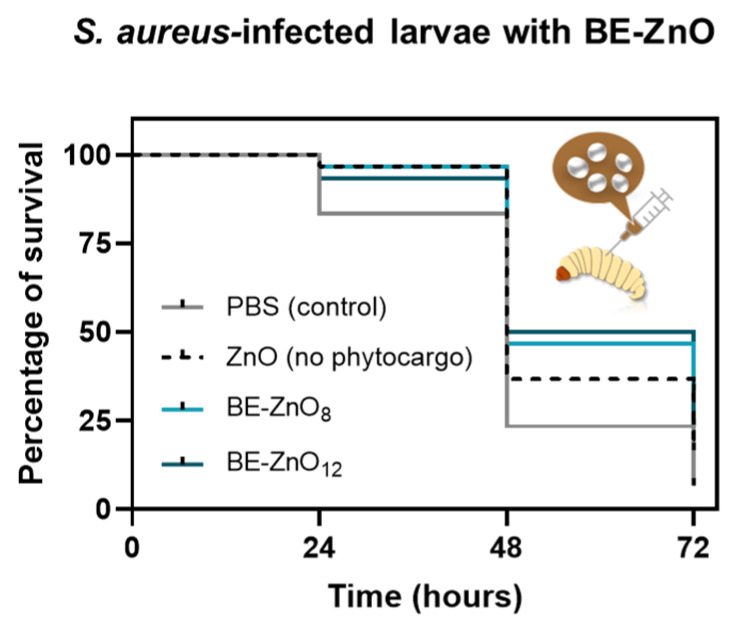
Antibacterial activity of ZnO particles without or with different phytocargos in larvae infected with *Staphylococcus aureus*. Survival curve of larvae treated with ZnO particles (100 mg/kg of larvae). Control groups were injected with PBS; each measurement represents the mean of at least three independent experiments.

**Figure 4 jfb-14-00463-f004:**
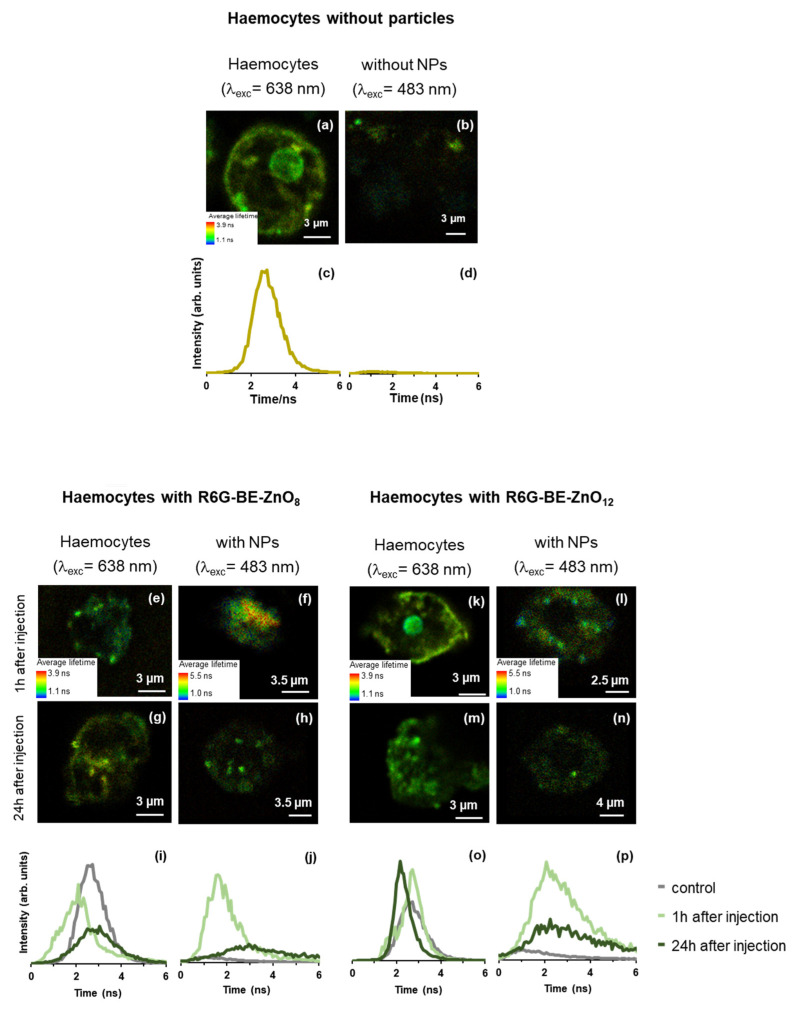
Characterisation of larvae haemocytes stained with Alexa 633 after injection with BE-ZnO NPs: FLIM images of haemocytes (**a**,**b**) without NPs acquired with λ_exc_ = 638 nm and λ_exc_ = 483 nm and (**c**,**d**) corresponding fluorescence lifetime distribution; FLIM images of haemocytes (**e**,**f**) 1 h and (**g**,**h**) 24 h after injection with BE-ZnO_8_ NPs acquired with λ_exc_ = 638 nm and λ_exc_ = 483 nm and (**i**,**j**) corresponding fluorescence lifetime distribution; FLIM images of haemocytes (**k**,**l**) 1 h and (**m**,**n**) 24 h after injection with BE-ZnO_12_ NPs acquired with λ_exc_ = 638 nm and λ_exc_ = 483 nm and (**o**,**p**) corresponding fluorescence lifetime distribution.

**Figure 5 jfb-14-00463-f005:**
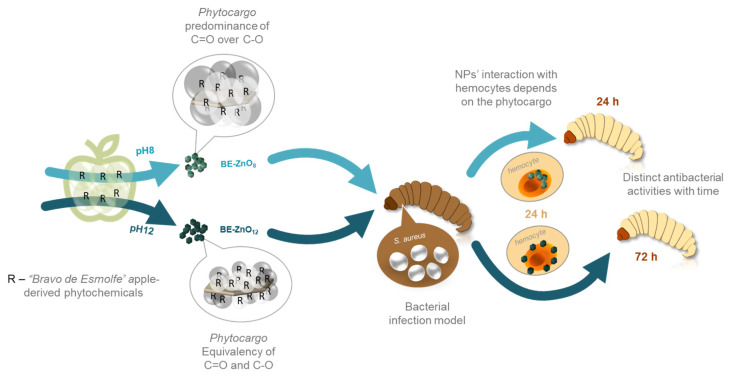
Scheme for the interaction of BE-ZnO NPs harbouring distinct phytocargos with larvae haemocytes and their antibacterial activity.

**Figure 6 jfb-14-00463-f006:**
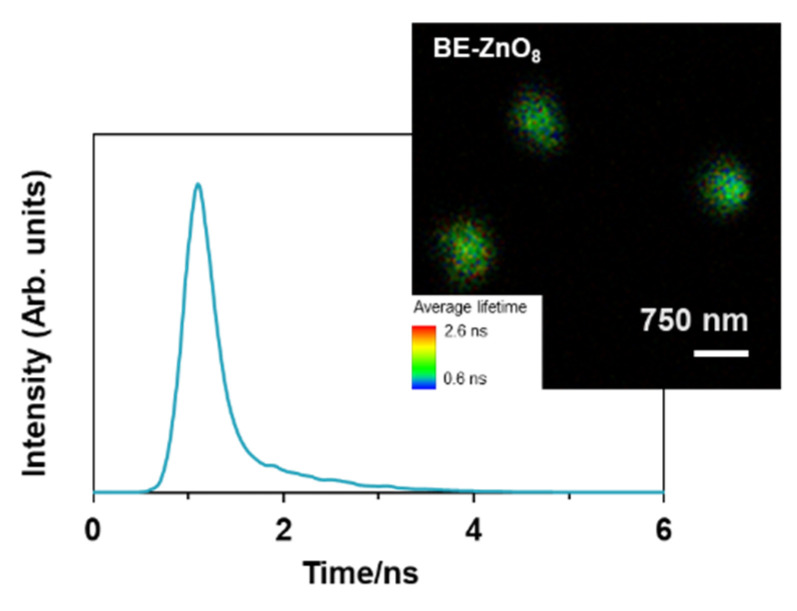
Photoluminescence characterisation of BE-ZnO_8_: fluorescence lifetime imaging images (FLIM) and associated fluorescence lifetime distribution λ_exc_ = 405 nm.

**Figure 7 jfb-14-00463-f007:**
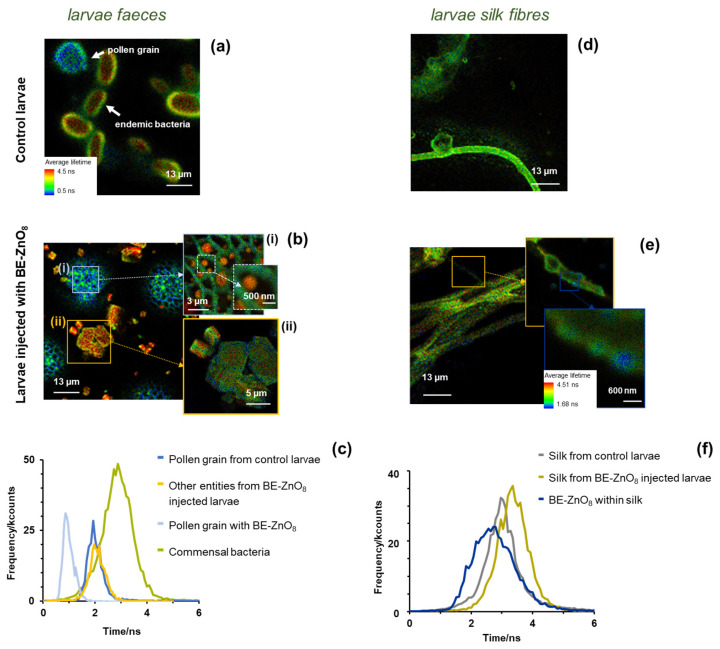
FLIM analyses (**a**–**c**) of larvae faeces and (**d**–**f**) silk fibres collected 48 h after injection with BE-ZnO_8_ NPs (200 mg/kg) or PBS (control); (**a**,**b**) FLIM images of faeces in the absence and presence of BE-ZnO_8_ NPs, respectively, with magnified insets marked as (i) and (ii), and (**c**) corresponding fluorescence lifetime distribution; (**d**,**e**) FLIM images of silk fibres in the absence and presence of BE-ZnO_8_ NPs, respectively, and (**f**) corresponding fluorescence lifetime distribution. FLIM analyses were made at λ_exc_ = 405 nm.

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
