# Peer review of "Antibacterial Activity of ZnO Nanoparticles in a Staphylococcus-aureus-Infected Galleria mellonella Model Is Tuned by Different Apple-Derived Phytocargos"

_jfb, 2023, doi:10.3390/jfb14090463_

Round 1

Reviewer 1 Report

Please follow instruction given in Word file of reviewed manuscript.

1. When you are writing scientific work you must be precise. Concretely in this manuscript, you can not come with general conclusion. Please come only with conclusion that are supported with experimental results performed and write them precisely.
2. I do not see why supplementary file is not part of the manuscript, please try to merge them.

Who wrote manuscript have improve English vocabulary. 

Once again, you have to express yourself precisely. 

Reviewer 2 Report

The manuscript “Antibacterial activity of ZnO in infected Galleria mellonella model is tuned by different apple phytocargos” presents a quite solid study on the green-synthesised metal oxide nanoparticles. Generally, it can be published in the J. Funct. Biomater. as it is within the scope of the journal and can attract some interest of the readers. The manuscript is written in a good English, it’s logically organized, the Conclusions are supported with the experimental results.

Nevertheless, the paper has one serious problem. And some minor ones. I hope, the authors will accept a challenge of major revision. Let’s go!

The main disadvantage of the work is the absence of proper controls, I mean the unloaded ZnO NPs synthesised under the same conditions. Without their characterization, it is hard to discuss the reasons for different activity and loading of the loaded samples.

The similar note is on the pure plant extract control. Why it was not used to compare the data on the extract-loaded NPs?

I am a material scientist and a chemist, I am not sure on the accuracy of the use of 10 units in each experimental group. Can groups of 10 ensure adequate statistics?

From the chemical point of view, the hydrolysis of C=O groups to produce C–O under ambient conditions seems incorrect. I recommend consulting organic chemists to avoid incorrect statements. The same is for the claim “pH increase results in C=O hydrolysing into carboxylic acids”. Carboxylic acids do contain C=O groups.

The second chemical comment arises from the mechanism of ZnO formation under different pH. Are the authors sure that the chemical composition of the ZnO samples is identical? I mean the OH and carbonate content in the ZnO particles prepared using simple treatment with alkali. The carbonate content in ZnO can add to the intensity of XPS peaks. Note, the XRD data for the samples differ. The difference is quite small, but it can be crucial for the property.

The content and the composition of the organic matter in the composite nanoparticles should be estimated even semiquantatively. By the way, Table S1 presents not EDX but XPS results. I think that the comparison of the FTIR data for the extract and the extract-loaded NPs is needed. What exact organics adsorb on the NPs surface? Generally, the discussion of the organic content basing on only XPS seems quite unfounded.

English is quite good.

Round 2

Reviewer 2 Report

I read the revised version of the manuscript ans the authors' answers with a great disappointment. It seems to me that virtually all of my concerns were ignored. 

Below, I will address the authors’ comments to my concerns.

Concern 1: the absence of proper controls, the unloaded ZnO NPs.

Answer 1: When synthesizing the particles using the exact same procedure (without extract), the resulting structures turned out to be micrometric flowers. We decided against including these micrometric particles in the study.

Comment 1: I wonder why even a short discussion of this experimental fact was not included in a revised version of the manuscript?

Concern 2: The absence of pure plant extract control to compare with the extract-loaded NPs.

Answer 2: a conclusive comparison is difficult as it is hard to distinguish individual peaks and accurately assign them to specific substances

Comment 2: Again, why not to compare? From the data in the answers I see that plant extract peaks (there is no need to accurately assign the peaks, even qualitative comparison is quite enough) are absent in extract-loaded NPs. Is it true? Why? Why this fact is obscured from the reader?

Concern 3: accuracy of the use of 10 units in each experimental group

Answer 3: 10 larvae should be utilized ethically

Comment 3: Ok.

Concern 4: the hydrolysis of C=O groups to produce C–O under ambient conditions seems incorrect

Answer 4: Revised.

Comment: Ok.

Concern 4: mechanism of ZnO formation under different pH, carbonate content in ZnO

Answer 4.1: There is no evidence of carbonates present in the samples.

Comment 4.1: Why do the authors think so?

Answer 4.2: we can confirm the presence of materials other than ZnO, as supported by SAED and XRD analysis.

Comment 4.2: I did not catch the idea… there ARE materials other than ZnO or not?

Answer 4.3: different profiles can be assigned to each of the nanoparticles, indicating that the XPS data indeed originates from distinct phytocargos rather than the precipitation of compounds other than ZnO.

Comment 4.3: I did not catch the idea.

Concern 5: content and the composition of the organic matter; the discussion of the organic content basing on only XPS seems quite unfounded.

Answer 5: identification of the exact organics is a challenging task

Comment 5: Again, the estimation of organics can be made by alternative methods, XPS is not the method of choice to analyse quantitatively the organic content.

As the authors did not tried to address my comments I feel that the manuscript should be rejected.

Acceptable

Author Response

>> The authors express their gratitude to the Editor and the reviewers for dedicating their time once more to reviewing our manuscript.

>> To distinguish the recent modifications, all newly added content in the manuscript has been highlighted in blue, while the previous changes remain marked in yellow.

>> The authors strongly believe that the manuscript has been improved further with the Editor and reviewers' valuable comments, and it now complies with the high standards of the journal. We trust that the manuscript is now ready for publication.

Reviewer#2

I read the revised version of the manuscript ans the authors' answers with a great disappointment. It seems to me that virtually all of my concerns were ignored.

>> There might have been some misunderstanding, as the comments given are indeed relevant and were already a concern for the authors. They were addressed carefully in the previous review round. The authors sincerely regret any disappointment experienced by the reviewer. In this current round, the authors are committed to making every effort to meet the reviewer's expectations.

Concern 1: the absence of proper controls, the unloaded ZnO NPs.

Answer 1: When synthesizing the particles using the exact same procedure (without extract), the resulting structures turned out to be micrometric flowers. We decided against including these micrometric particles in the study.

Comment 1: I wonder why even a short discussion of this experimental fact was not included in a revised version of the manuscript?

>> Information relating to the unloaded ZnO NPs was added both in the SI and in the main manuscript. Since the physicochemical characterization of the particles synthesized with no phytocargo has been already reported by the authors [R1, R2], the morphological features were added in SI, whereas the novelty relating to their bioactivity (larvae health index, MIC, MBC and antibacterial efficacy in the infected model) were added in the main manuscript.

[R1] Alves, M.M.; Andrade, S.M.; Grenho, L.; Fernandes, M.H.; Santos, C.; Montemor, M.F. Influence of apple phytochemicals in ZnO nanoparticles formation, photoluminescence and biocompatibility for biomedical applications. Materials Science and Engineering: C 2019, 101, 76-87, doi:https://doi.org/10.1016/j.msec.2019.03.084

[R2] Soliman, M.M.A.; Alegria, E.C.B.A.; Ribeiro, A.P.C.; Alves, M.M.; Saraiva, M.S.; Fátima Montemor, M.; Pombeiro, A.J.L. Green synthesis of zinc oxide particles with apple-derived compounds and their application as catalysts in the transesterification of methyl benzoates. Dalton Transactions 2020, 49, 6488-6494, doi:10.1039/d0dt01069c

Concern 2: The absence of pure plant extract control to compare with the extract-loaded NPs.

Answer 2: a conclusive comparison is difficult as it is hard to distinguish individual peaks and accurately assign them to specific substances

Comment 2: Again, why not to compare? From the data in the answers I see that plant extract peaks (there is no need to accurately assign the peaks, even qualitative comparison is quite enough) are absent in extract-loaded NPs. Is it true? Why? Why this fact is obscured from the reader?

>> The discussion relating to ATR from the apple extract and BE-ZnO NPs was included in the main manuscript, and the spectra were added in SI along with the corresponding materials and methods.

Concern 4: mechanism of ZnO formation under different pH, carbonate content in ZnO

Answer 4.1: There is no evidence of carbonates present in the samples.

Comment 4.1: Why do the authors think so?

>> Only morphologies typically associated with ZnO, such as hexagonal shapes have been observed. There is no evidence of morphological (round to lamina-like particles) nor crystalline information indicating the presence of zinc carbonates. Consequently, it can be inferred that, at least within the detection limit of the characterization techniques used, the samples are composed of pure ZnO NPs. Also, following the literature, the green-synthetic procedure employed leads to the formation of pure ZnO phases [R3].

[R3] Bandeira, M.; Giovanela, M.; Roesch-Ely, M.; Devine, D.M.; da Silva Crespo, J. Green synthesis of zinc oxide nanoparticles: A review of the synthesis methodology and mechanism of formation. Sustainable Chemistry and Pharmacy 2020, 15, 100223, doi:https://doi.org/10.1016/j.scp.2020.100223

Answer 4.2: we can confirm the presence of materials other than ZnO, as supported by SAED and XRD analysis.

Comment 4.2: I did not catch the idea… there ARE materials other than ZnO or not?

Answer 4.3: different profiles can be assigned to each of the nanoparticles, indicating that the XPS data indeed originates from distinct phytocargos rather than the precipitation of compounds other than ZnO.

Comment 4.3: I did not catch the idea.

>> We apologize for any confusion caused by these sentences. To clarify, we have not identified any material other than ZnO in our study. Moreover, the data collected by XPS is related to the phytochemicals present in the NPs. There is no evidence of zinc carbonate formation, i.e. the XPS data from Zn 2p has lower energies than those reported for zinc carbonates [R4]. The concern of the reviewer, i.e. possible formation of carbonates, has been incorporated into the manuscript aiming to provide evidence to readers who may share the same concern.

[R4] Liu, Z.; Teng, F. Understanding the Correlation of Crystal Atoms with Photochemistry Property: Zn5(OH)6(CO3)2 vs. ZnCO3. ChemistrySelect 2018, 3, 8886-8894, doi:https://doi.org/10.1002/slct.201801420

Concern 5: content and the composition of the organic matter; the discussion of the organic content basing on only XPS seems quite unfounded.

Answer 5: identification of the exact organics is a challenging task

Comment 5: Again, the estimation of organics can be made by alternative methods, XPS is not the method of choice to analyse quantitatively the organic content.

>> Apart from the ATR data added, we are not clear about the reviewer's thoughts about the method of choice to ‘analyze quantitatively the organic content’. As such, we will reveal our thoughts on this comment. The ICP technique could be a consideration, providing precise C and O measurements. However, the conclusions drawn would not change, i.e. variations in pH led to distinct phytocargos, as asserted by EDS measurements (Table S1). The central issue lies in determining the nature of these phytocargos to comprehend the resulting bioactivities, which cannot be fully revealed by the raw C and O content. HPLC techniques, while powerful, cannot be considered an option due to particle clogging within the columns. An acidic dissolution could present an alternative; however, the potential disparity in phytochemical modifications compared to those within the NPs is a concern. Thus, the XPS appears as the technique that can address this critical aspect.

As the authors did not tried to address my comments I feel that the manuscript should be rejected.

>> We sincerely hope that this time we have been able to meet the reviewer’s expectations.